# Gender-Based Socio-Economic Inequalities in the Pre-Vaccination Era of the COVID-19 Pandemic in Istanbul: A Neighborhood-Level Analysis of Excess Mortality

**DOI:** 10.3390/healthcare12141406

**Published:** 2024-07-15

**Authors:** İlker Kayı, Mehmet Gönen, Sibel Sakarya, Önder Yüksel Eryiğit, Önder Ergönül

**Affiliations:** 1Department of Public Health, School of Medicine, Koc University, Istanbul 34010, Türkiye; ssakarya@ku.edu.tr; 2Department of Global Health, Graduate School of Health Sciences, Koc University, Istanbul 34450, Türkiye; 3Department of Industrial Engineering, College of Engineering, Koc University, Istanbul 34450, Türkiye; mehmetgonen@ku.edu.tr; 4Koc University Is Bank Research Center for Infectious Diseases, Koc University Hospital, Istanbul 34010, Türkiye; oergonul@ku.edu.tr; 5Health Department, Istanbul Metropolitan Municipality, Istanbul 34134, Türkiye; onder.eryigit@ibb.gov.tr; 6Department of Infectious Diseases and Clinical Microbiology, School of Medicine, Koc University, Istanbul 34450, Turkey

**Keywords:** spatial disparities, socio-economic disadvantage, gender, all-cause excess mortality, ecological study

## Abstract

Worldwide excess mortality (EM) data have the potential to provide a better estimation of the impact of the pandemic. This study aims to investigate and map the inequalities in EM in Istanbul during the pre-vaccination era of the COVID-19 pandemic in 2020 and its association with selected demographic and socio-economic variables at the neighborhood level according to gender. This ecological study was conducted with the EM data of Istanbul. The EM data were obtained from the Istanbul Metropolitan Municipality (IMM) and analyzed according to socio-demographic indicators (gender, age), neighborhood-level indicators (population density, educational attainment) and neighborhood vulnerability (socio-economic and transportation) for the 808 neighborhoods, then presented separately according to gender to examine gender-specific factors. Socio-economic and transportation vulnerability indexes are provided the IMM. The excess mortality rate per 1000 (EMR) in 2020 has been calculated by using the number of deaths in the years 2018–2019. We have mapped EMRs of each neighborhood and used linear regression analysis in three datasets to examine gender specific factors. EMRs in Istanbul showed two peaks one in April and one in November. Male EMRs were higher compared to females in Istanbul during the pre-vaccination era of the pandemic. Higher EMRs were observed in neighborhoods with a higher share of 50+ year old age groups and higher neighborhood socio-economic vulnerability scores. Neighborhood socio-economic vulnerability was significantly associated with EMRs in males but not in females. Unequal distribution of EM between neighborhoods underlines the need for gender-specific pandemic measures to alleviate the burden of the COVID-19 pandemic, especially in socio-economically vulnerable settings. Increased use of area-based indicators with a gender perspective can enhance pandemic measures.

## 1. Introduction

Starting in early 2020, the COVID-19 pandemic, caused by the novel virus SARS-CoV-2, has been difficult to manage due to many unknown factors regarding the pathogen that have led to more than 6.5 million deaths globally [1]. COVID-19 mortality rates have often been criticized to be an underestimation due to unstandardized operational procedures for the identification of the cases and causes of death in regard to confirming COVID-19. With the World Health Organization’s guidance, the number of confirmed cases and mortality numbers reported by each country since the beginning of the pandemic show a heterogeneity due to misclassifications of the cases because of the varying testing and diagnostic criteria, the percentage of false-negative results based on test kit quality, misclassification of clinically diagnosed cases, and errors in certification of cause of death [2]. Therefore, excess mortality (EM) can be used as a more reliable indicator of the impact rather than COVID-specific mortality data [3]. EM data include deaths directly attributed to COVID-19 along with the ones that are indirectly related with the pandemic, such as deaths due to delayed health care seeking behavior, cancellation of regular health services and increased mental health problems that lead to suicidal ideation [4]. The EM data provide an unbiased estimation of the impact of the public health emergencies, allowing for international comparisons. Analysis of the EM data has the potential to provide more complete and timely information on the true burden of the COVID-19 pandemic [5].

At end of December 2020, Turkey was the seventh country with highest cumulative case numbers of COVID-19, with 2.1 million confirmed cases. However, when compared to cumulative death numbers, Turkey ranks 18th in the world, with 20,388 deaths due to COVID-19 [1]. The analysis provided by COVID-19 Excess Mortality Collaborators based on all-cause mortality reports from 74 countries and territories and 266 subnational location revealed that the real impact of the pandemic was much greater than the deaths reported only due to COVID-19 [3]. According to the study results, although 5.94 million COVID-19 deaths were reported in the world between 2020 and 2021, 18.2 million (95% uncertainty interval 17.1–19.6) people died worldwide due to the COVID-19 pandemic as measured by EM. Estimates similarly showed that, for Turkey, the ratio of EM to reported COVID-19 mortality was around two to five [3].

Istanbul—the13th most crowded city in the world with its 15,840,900 inhabitants– became the epicenter of the pandemic in Turkey since the beginning, with higher COVID-19 cases and mortality rates. It is no surprise, because Istanbul is one of the international hubs of the world for transit flights and business, and, locally, 18.7% of the population in Turkey live in Istanbul, making it the most inhabited city nationally [6]. The official and excess deaths in Istanbul were 24,218 and 37,514 [7] in 2020. Limited studies have illustrated the association of socio-economic status with EM at the district level [8]; however, these studies lack age and gender variables.

It is well established that gender and socio-economic status played a significant role in the unequal distribution of EM during the pandemic. Biomedical factors were shown to be associated with the mortality rates that were higher in males, the elderly population, and among those living with chronic conditions such as hypertension, diabetes, and heart disease [9]. Along with these biomedical risk factors, evidence on the socio-economic and environmental factors show that the COVID-19 pandemic affected vulnerable groups in many societies [10,11]. Disparities in COVID-19 death rates due to conditions beyond medical reasons were studied in several other studies. A study from the United States of America (USA) highlighted the interconnectedness of racial, socio-economic and gender inequality in shaping the death rates during the pandemic [12]. Stokes et al. illustrated how variations in sociodemographic and health characteristics at the county level significantly lead to inequalities in excess mortality due to COVID-19 [13]. These findings emphasize the impact of local context, demographics and socio-economic status on exacerbating or mitigating the outcomes of the pandemic in the population.

Additionally, gender disparities in COVID-19 mortality are clearly demonstrated in the literature. According to Cifuentes et al., factors such as age and gender significantly influence the risk of COVID-19 mortality, which is higher among males [14]. Rizzi, Strozza and Zarulli’s analysis of sex differences in excess death risk during the first wave of the COVID-19 pandemic in Italy showed the existence of such a difference, but the results were nuanced and inconsistent [15]. Another study by Pothisiri et al., which observed fluctuations in excess all-cause mortality rates between genders during the pandemic, highlights the necessity to address gender-specific vulnerabilities due to the gender gap in mortality rates [16].

These findings underscore the importance of comprehensive inquiries of the mortality data that takes the intersecting factors into account for providing a better response to the impact of pandemics until vaccines are available to prevent pandemic-related mortality and morbidity. Previous studies have shown gender differences in mortality during the pre-vaccination era; however, gender-based risk factors are limited in the literature. This study will investigate such factors with a gender specific approach in Istanbul, Turkey, during the first and second waves of the pandemic in 2020 before the vaccines were made available for priority groups (beginning in January 2021) [17]. However, independent researchers’ access to the COVID-19-related surveillance data required for the task has been restricted in Turkey [18]. Also, publicly available information on COVID-19 in Turkey lacks the distribution of cases and mortality rates according to demographics, gender, and socio-economic level [19]. Studies on COVID-19 excess mortality from Turkey have been limited to cities with collaborative municipalities that share mortality data with researchers, especially Istanbul, where researchers can obtain data from the Istanbul Metropolitan Municipality (IMM) [7,8,19] along with recent socio-economic and transportation vulnerability indexes at the neighborhood level, which provide the opportunity to analyze the local context in a densely populated city. Therefore, in this ecological study, we aimed to investigate and map the inequalities in EM in Istanbul during the pre-vaccination era of the COVID-19 pandemic in 2020 and its association with selected demographic and socio-economic variables at the neighborhood level according to gender. The proposed hypothesis of our study were as follows:Older age groups and males are at higher risk of mortality during the pre-vaccination era of the pandemic.Neighborhood socio-economic vulnerability is associated with increased excess mortality rates during the pre-vaccination era of the COVID-19 pandemic.Neighborhood socio-economic vulnerability influences excess mortality rates differently according to gender during the pre-vaccination era of the COVID-19 pandemic.

## 2. Materials and Methods

### 2.1. Data Source

This study adopts an ecological approach by using the death numbers and regarding demographic information such as gender, age and neighborhood provided by the IMM, which regulates the funeral referral services by collecting the death certificates of deceased persons [8]. The IMM Directorate of Cemeteries holds a record of the death certificates that are issued by a medical doctor on a centralized death reporting system. The death report is sent to the IMM for organizing the funeral and burial services, followed by the Turkish Statistical Institute for recording the mortality data. Mortality data used in this study were obtained from IMM with the following information of the deceased persons: age, gender, residential neighborhood and time of death. Death of persons who are not registered residents of Istanbul but deceased in Istanbul are not included in the dataset. We obtained the data on neighborhood-level socio-economic and transportation vulnerability scores from the IMM based on the work by the Statistical Office in the municipality [20]. We also used a neighborhood-level digital map of Istanbul containing neighborhood-level data, including the population censuses, population density, land size of the neighborhoods, age groups, educational attainment and gender composition according to the most recent Turkish Statistical Institute Data for each variable.

### 2.2. Preparation of the Dataset for Analysis

The variables in the initial dataset obtained from the IMM were categorized based on neighborhoods to calculate the monthly deaths in each neighborhood from 2018 to 2020, as the unit of analysis in our study were neighborhood-level indicators. In total we have used three datasets for total, male and female deaths each consisting of neighborhood-level variables brought together. The dataset for total deaths included gender as a variable, which was absent in other datasets. Neighborhood-level variables used in the datasets are provided below.

### 2.3. Variable Definition

The variables used in the study are derived from neighborhood-level data, including population density, share of 50+ years old residents, educational attainment, socio-economic and transportation vulnerability indexes and excess mortality rate (EMR) per 1000 population. The required data regarding the variables, namely population censuses, share of 50+ years old residents and educational attainment at the neighborhood level, have been obtained from the Turkish Statistical Institute open data source [6] and were embedded in the digital map tool along with the size of the land area for each neighborhood. Definition of each variable used in the study are presented below.

Independent variables:Gender data was obtained from the IMM Statistical Office in the initial dataset, which reported it as binary variable (male and female).Time of death was reported by the IMM as the day, month and year of death for the years from 2018 to 2020.Neighborhoods of deceased people were obtained from the IMM for each individual.Population density has been calculated for each neighborhood as number of people per kilometer square. The population census of 2020 and land sizes are obtained from the digital map that contained the Turkish Statistical Institute’s latest data, as mentioned above.Share of 50+ year old residents indicated the percentage of people living in each neighborhood who were above 50 years old. Age distribution of each neighborhood was included in the digital map used in the study.Educational attainment was categorized as percentages of people in each neighborhood with less than a high school degree, high school degree and college or university degree (undergraduate or graduate degrees were combined). Categories of educational degrees (elementary, secondary, high school, college and university) in percentages for each neighborhood were included in the digital map used in the study.The Socio-Economic Vulnerability Index (SEVI) is a composite scale developed by IMM to measure neighborhood-level socio-economic disadvantage. The IMM Statistical Office used age dependency ratio, ratio of working population to dependent population, ratio of university graduates, household size, number of households applied to social support services, number of banks, rental prices for housing and income levels at each neighborhood to formulate the index.The Transportation Vulnerability Index (TVI) is a composite scale developed by IMM to measure the neighborhood-level disadvantage in regard to the transportation services available. The TVI includes number of travels, share of public transportation in vehicle transportation, passenger density per station, number of passengers with disabilities, and the number of passengers over 65 years of age at the neighborhood level.

The SEVI and TVI data for each neighborhood were obtained from the IMM Statistical Office, which adopted a methodology by utilizing a variety of data to develop a composite score. The IMM Statistical Office has standardized and then transformed the primary and secondary sources of data into a unidimensional score format by applying datamining techniques. The standardized data were converted to normalized values for easier reading of the index. Thus, all values were positioned between 0 and 100 so that higher scores indicate increased vulnerability or disadvantage [20].

Dependent variables:EMRs at the neighborhood level constitute the dependent variable of the study. The mortality data included death records from 2018, 2019 and 2020. The number of excess deaths has been calculated for the year 2020 by extracting the number of deaths in a neighborhood in 2020 from the expected number of deaths calculated as the mean deaths of 2018 and 2019 (#deaths in 2018 + #deaths in 2019 divided by 2) of the same neighborhoods. The EMR was calculated by dividing the number of excess deaths in each neighborhood by the total neighborhood population in 2020 (per 1000). This procedure was repeated for male and female deaths with a denominator of male or female population in each neighborhood. Positive results indicated that there was an increase in the death rate in 2020 compared to 2018 and 2019, while negative results indicated a decrease in mortality.

### 2.4. Mapping EMRs of Istanbul Neighborhoods

We used maps and map-tools packages in R for mapping the EMRs of each neighborhood. As the EMR could take a negative value due to decreased mortality, we utilized the heatmap function that visualized negative values (decreased EMR) in blue and positive values (increased EMR) in red. Darker colors indicated higher increases or decreases.

### 2.5. Statistical Analysis

The analysis was conducted for 808 neighborhoods out of 961 in 39 districts of Istanbul. The neighborhoods with less than a 1000 population size have been excluded as they caused an error in the variance of EMRs. Temporality of the EMRs was presented as the monthly distribution of the EMRs from January to December for the years 2018, 2019 and 2020 according to gender. We used geographical information systems to color map the EMRs. Descriptive statistics regarding the neighborhoods included population density per one kilometer square, percentage of the population in each neighborhood with certain levels of educational attainment, percentage of people over 50 years old in each neighborhood, the SEVI and TVI scores of each neighborhood and the gender-based EMRs. We have presented a mean, median and interquartile range of each variable. For regression analysis, we have used the EMRs per 1000 people as the dependent variable of the simple and multivariable linear regression models to analyze the association of EMRs with sociodemographic variables and vulnerability indexes. All analysis were performed in the R Studio. All variables except educational attainment, due to its coverage under SEVI, have been included in the multivariable model. Statistical significance was accepted as *p* < 0.05. The Social Sciences and Humanities Ethics Board of Koc University Committee on Human Research approved the study (Approval Number: 2023.134.IRB3.057).

## 3. Results

The total number of deaths in Istanbul in 2018, 2019 and 2020 were 52,342, 53,937, and 67,977, respectively. The total number of excess deaths was 14,838 in 2020 in Istanbul during the first and second waves of the pandemic. Figure 1 depicts the temporal distribution of all-cause death numbers. There have been two peaks observed in the death numbers. The first one was in April 2020 and second was in November 2020. The increasing pattern of the changes in death numbers indicated a seasonality, as the beginning of the first peak was in February and the second peak was in September.

Table 1 shows the distribution of sociodemographic variables and EMR per 1000 capita for 808 neighborhoods of 39 districts in Istanbul with an over-1000 population. The EMR has a range between −6.88 and 8.23. Also, the difference in median EMR between the 10 highest (least vulnerable) and 10 lowest (most vulnerable) neighborhoods is 1.05. Figure 2 illustrates the mapping of EMR and other covariates in the 808 neighborhoods of Istanbul included in our analyses.

The simple linear regression analysis of the association between covariates and EMR has been shown in Table 2. In total, a higher percentage of population being over 50 years of age (*p* = 0.002), a higher percentage of people with under a high school degree (*p* = 0.009), higher levels of the SEVI score (*p* = 0.014) and higher levels of the TVI score (*p* = 0.007) have been found to be associated with higher EMRs. Among females, the simple linear regression analysis of the association between the covariates and EMR showed that neighborhoods with a higher percentage of people over 50 years of age had a higher EMR (*p* = 0.045). However, according to the results regarding the association of sociodemographic covariates with EMRs among males, higher percentage of population over 50 years of age (*p* = 0.014), higher percentage of people under high school degree (*p* = 0.002), higher levels of SEVI score (*p* = 0.005), and higher levels of TVI score (*p* = 0.021) have been found to be associated with higher levels of EMR.

The multivariable linear regression analysis of the association between selected covariates and EMR is shown in Table 3. For the whole population, higher neighborhood EMR was associated with higher percentage of people over 50 years of age (*p* < 0.001) and higher neighborhood SEVI score (*p* < 0.001). While female EMR was only associated with higher percentage of people over 50 years of age (*p* = 0.015), male EMR was associated with higher population density (*p* = 0.027), higher percentage of people over 50 years of age (*p* < 0.001) and higher neighborhood SEVI score (*p* < 0.001).

## 4. Discussion

Our findings presented in Box 1 indicate that there was an increase in the number of deaths in Istanbul above the expected amount in the pre-vaccination era of the COVID-19 pandemic, which can be attributed to the direct effects of the disease and indirect effects of the pandemic conditions in the absence of any other major explanatory health-related event. We have observed differences based on gender, and area-based vulnerabilities regarding socio-economic status and transportation in EMRs at the neighborhood level. The multivariable linear regression showed that the total EMR was associated with higher socio-economic vulnerability and a higher share of 50+ year old residents but not with transportation vulnerability.

Box 1Key findings.The temporal distribution of total EMRs in Istanbul mirrored two peaks throughout the pre-vaccination era observed in April and November.Male EMRs were higher compared to females in Istanbul during the pre-vaccination era of the pandemic, with notable increases seen in both peaks of the EMR surges.Age significantly influenced EMRs during the pandemic, with higher mortality seen in neighborhoods with a higher share of 50+ year old age groups.The distribution of EM during the pandemic was uneven across neighborhoods, with higher neighborhood socio-economic vulnerability associated with increased total EMRs.Neighborhood socio-economic vulnerability was significantly associated with EMRs in males but not in females, indicating a gender-specific impact of neighborhood socio-economic vulnerability on mortality rates.

The temporal distribution of total EMR in the pre-vaccination era of the pandemic has been similar to other countries in the northern hemisphere, such as Jordan [21] and the United States of America (USA) [22]. Our findings indicated that there was an increase in EMRs in Istanbul since March, making its first peak in April and second peak in November 2020. Several reports also reported that the EMRs increased in mid-March in 2020 in Turkey [7,8,19]. According to these accounts, while the number of excess deaths already was around two thousand, on March 11, 2020, the MoH declared the first official COVID-19 case in Turkey that was detected in Istanbul [18]. Considering the absence of other health emergencies to explain the increase in the EMR during the time, it would be fair to assume a delay in the detection of COVID-19 cases since the beginning of the virus spread in Turkey due to the testing strategies of the MoH, which lagged behind the World Organization’s recommendation of five per one thousand person per day in 2020 and 2021 [17].

Another finding of our study was that the male EMR was higher compared to females in Istanbul in the pre-vaccination era of the pandemic, as can be seen in both peaks of the EMR increases. Similar findings have been shown in Jordan [21], England [23], and Philadelphia, USA [22]. In another study comparing morality trends of 37 countries during the pandemic, life expectancy was found to decline in men more than women, except in Luxembourg [24]. Apart from the general mortality trend, the most plausible explanation for the difference in male and female mortality might be the high case fatality rates (CFRs) of COVID-19 in men [25]. Rizzi, Strozza and Zarulli argue that, in order for CFRs to have an influence in the excess mortality observed in a year compared to previous years, the health event in that particular year must be substantial to link the excess deaths with the cause of death, otherwise CFRs are negligible [15]. We now know from the scale and the worldwide impact of the COVID-19 that it is the only explanatory factor of thess excess deaths. Whether this difference is based on sex or gender is beyond the scope of our study; however, it has been well established in the literature that men have higher mortality rates compared to women due to biomedical factors, gender roles, socio-economic influences (which have different consequences in terms of behavioral choices) and exposure to environmental and work-related risk factors [26].

During the two peaks of the excess mortality throughout 2020, the EMRs were also unevenly distributed based on age. According to our findings, neighborhoods with a share of 50+ year-old residents had significantly higher total EMRs, and the association continued to be significant separately both for male and female EMRs. Nevertheless, beyond the number of deaths and EMRs which were higher for males, there was another disparity that can be observed in our findings based on gender. While the EMR was nearly 2.5 times higher in female deaths for those neighborhoods with a higher share of 50+ residents, it was 4.5 times increased in male deaths. Similar findings have been reported during the early phases of the pandemic from Italy [15], Sweden [27], and England [28]. An association between increased age and EMR is an expected phenomenon; however, studies vary in terms of the cut-off point. The association between age and EM during the pandemic was shown to be significant for 60+ year-old people in Jordan, which was explained by the reduction in common death causes such as traffic accidents due to lock downs and travel restrictions [21]. The EMRs during the pandemic in Philadelphia were significantly elevated for men and women over 40 years old due to increased early age mortality among racial minority groups [22]. It has also been demonstrated that EM was higher in the oldest age groups in an analysis of 20 countries except for Peru, where it was also higher in 45-year-old and younger groups [29]. According to a national cohort study in Turkey on risk factors of non-communicable diseases, the prevalence of multimorbidity was 2% under the age of 45, but it increased to 20% in the 45–54 year age group, and continued to increase with age [30]. Our age cut-off was at 50 years; therefore, it corresponds to the ages with a high prevalence of chronic conditions, which might provide an explanation for the association between the share of 50+ year old residents in neighborhoods in Istanbul and EMR.

Apart from the temporal sequence and age distribution of deaths, we have observed area-based differences in the distribution of EMRs. Disadvantaged neighborhoods in terms of socio-economic development and transportation services had a higher share of EMRs. However, when we have conducted the analysis according to gender, the neighborhood TVI score was not associated with female EMRs but male EMRs. This might be due to different transportation patterns between genders, as shown by other studies [31,32]. Nevertheless, when analyzed in a multivariable model, the significance of transportation vulnerability diminished, probably due to the collinearity with socio-economic vulnerability. In other words, dimensions of TVI, such as the number of vehicle transportations and transportation density, might also be a representation of socio-economic vulnerability. Similarly, the significant association in the simple regression analysis between educational degree and total EMR diminished for female deaths but persisted for male deaths when we made our gender-based analysis. The same trend continued in multivariable linear regression for neighborhood socio-economic vulnerability, which was significantly associated with only male deaths.

Our analysis based on the SEVI illustrates that distribution of EM during the pandemic was not even across neighborhoods and that higher neighborhood socio-economic vulnerability was associated with increased EMRs. Similarly, anomalies in excess deaths in Santiago, Chile, mostly occurred in municipalities with a low socio-economic status [33]. A neighborhood-level analysis of EM in Stockholm found that the EM was two times higher in socio-economically deprived areas [34]. Another study showed that the higher proportion of the county population living in disadvantaged areas had higher mortality rates in the USA [35]. The SEVI is a composite index derived from neighborhood-level indicators of education, employment, income, household size and deprivation. Therefore, it would not be wrong that the socio-economically disadvantaged neighborhoods of Istanbul are the places where people of lower socio-economic status reside. It has been well documented that COVID-19 mortality was associated with social determinants of health such as poverty, crowded households and lower education levels [36,37,38,39].

All around the world, while the upper class or white-collar workers enjoyed working from home, many essential workers in the healthcare, agriculture, security, critical manufacturing and services industries, that also contain a high number of lower-class jobs, were obliged to work onsite, which put an extra burden on their socio-economic disadvantage due to the increased likelihood of virus exposure [40]. Carranza et al. have showed that compliance to stay-at-home orders were much lower in the low-income zones of Santiago, Chile, due to lower class jobs that forced people to work during the sheltering periods [41]. When Turkey followed the course like other countries to enact stay-at-home orders, an exempt was provided for essential workforces. It is most likely that the neighborhoods of Istanbul with higher SEVI scores might be home to these essential workers with higher levels of exposure to the virus. According to a report by IMM, 20% of the lower socio-economic status household members continued to work during the pandemic at the same rate as pre-pandemic times [42]. Also, social and financial aid applicants during the pandemic were mostly workers (26.8%), unemployed persons (20.2%), and housewives (10.9%) who were residing in neighborhoods with lower SEVI scores in districts such as Esenyurt, Bagcilar, Kucukcekmece, Sultangazi and Umraniye, which had the higher number of applications [42]. Additionally, the SEVI includes household size in its composite nature. So, household sizes of these socio-economically disadvantaged neighborhoods of Istanbul might be higher than neighborhoods with low SEVI scores, which also might have contributed to the increased EMRs because of the increased likelihood of virus transmission. The same IMM report also indicates that 25% of the social and financial aid applicants had more than two children in the household [42].

One interesting finding of our study is that the significant association of neighborhood SEVI scores with EMRs diminishes for female deaths when we do our analysis separately according to gender. One reason for this finding might be the stark differences in workforce participation between men and women in Turkey. According to the Turkish Statistical Institute report for 2020, participation in the workforce was 67% for men compared to 30% for women [43]. It has been shown that working women had similar or increased death rates during the pandemic compared with men [44]. Therefore, it is more likely that men in socio-economically disadvantaged neighborhoods in Istanbul had higher participation in the workforce in the essential work categories, resulting in increased risk of exposure to the virus.

Also, another finding in our study was that there was a significant association between neighborhood population density and EMRs in males but not for females. Previous studies indicate that population density was significantly associated with all-cause mortality during the pandemic [23]. In their spatial analysis, Pilkington et al. found that densely populated areas with higher working-related mobility showed higher all-cause mortality during the pandemic in France [45]. In contexts where participation in the workforce showed stark differences according to gender, having to go to work from a densely populated neighborhood for men might have increased the likelihood of virus exposure and mortality in Istanbul.

This study has several important limitations that should be considered when interpreting the results. This is an ecological study that used neighborhoods as the unit of analysis. As such, the associations found between neighborhood-level socio-economic variables and EMRs do not necessarily indicate causal relationships at the individual level. Due to the no-data-sharing policy of the authorities, the study was unable to obtain data on both the individual level and cause-specific mortality to differentiate between COVID-19-related deaths and deaths from other causes that can lead to better causal inferences. Therefore, it is out of the scope of this study to determine how much of the excess mortality was directly attributable to COVID-19 versus other factors. Also, mortality datasets can have potential biases that may have affected the accuracy and completeness of the mortality data. Errors or inconsistencies in how the underlying cause of death is coded on death certificates could lead to misclassification of deaths. However, as we are aware of such conditions, we wanted to analyze all-cause-mortality data to overcome such biases to understand the true burden of the pandemic during the pre-vaccination era. Another problem with mortality datasets is that potential delays in registering deaths with the authorities during the pandemic could lead to an undercount of deaths, especially in the most recent time periods of the dataset. We believe this is a limited bias in our study, as we have obtained the data later than the pre-vaccination era in 2022. Lastly, as this study was conducted in Istanbul, Turkey, the findings may not be generalizable to other cities or countries with different socio-economic contexts and pandemic experiences. Replication of the study in other settings would help assess the external validity of the results. Despite these limitations, this study provides important insights into the socio-economic disparities in excess mortality based on gender during the COVID-19 pandemic in Istanbul. The findings highlight the need for more robust, timely and transparent mortality surveillance systems, as well as targeted public health interventions during health emergencies.

## 5. Conclusions

During the first year of the COVID-19 pandemic, when there was no vaccine available, our results illustrate that the burden of EM in Istanbul fell on the neighborhoods that had a higher proportion of 50+ year-olds in the population and higher socio-economic vulnerability scores. There was a gender differential in EMRs, with higher mortality rates being seen among men. However, gender-based analysis indicated neighborhood socio-economic vulnerability was only associated with male EMRs. Therefore, gender-based approaches are required to design preventive interventions for the socio-economically vulnerable settings in addition to uniform pandemic measures. However, there is still a need for detailed data to understand the individual level impact of the COVID-19 pandemic in Turkey.

## Figures and Tables

**Figure 1 healthcare-12-01406-f001:**
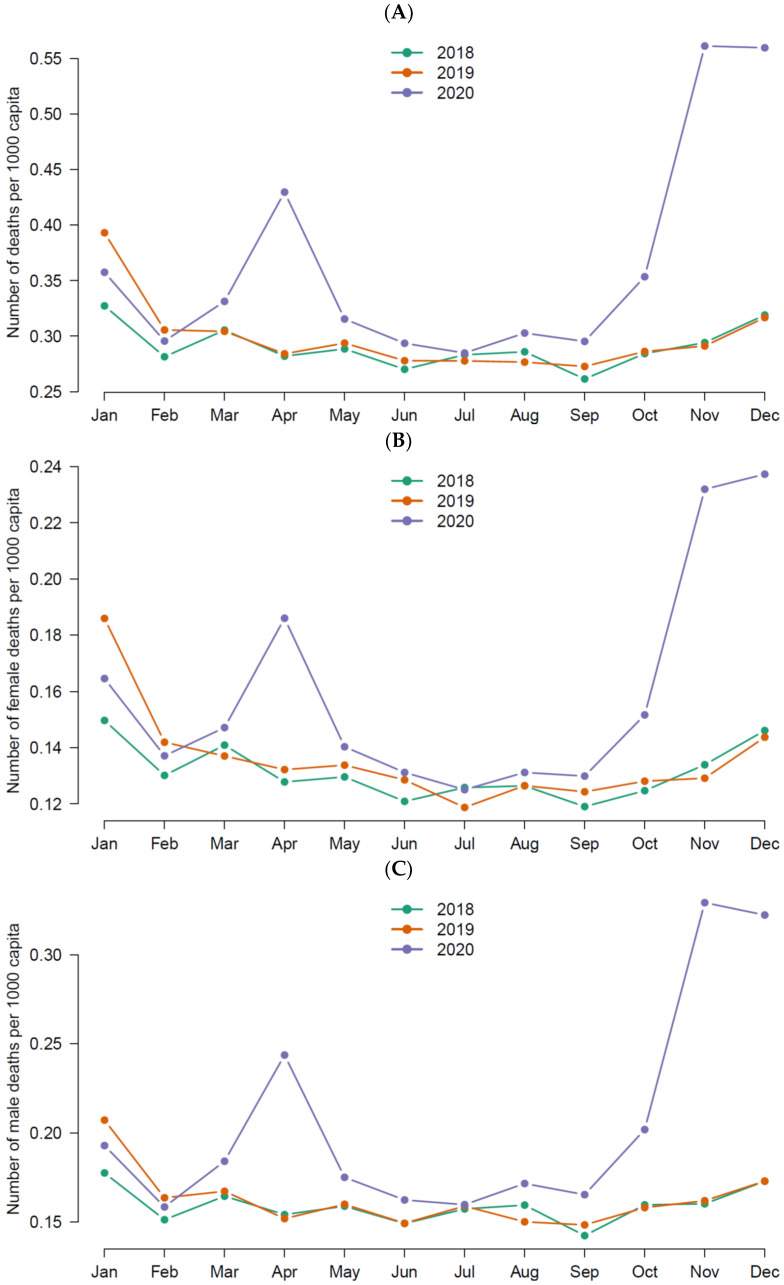
(**A**–**C**) Temporal (monthly) distribution number of deaths for the years 2018, 2019 and 2020 in Istanbul. Total (**A**); female (**B**); male (**C**).

**Figure 2 healthcare-12-01406-f002:**
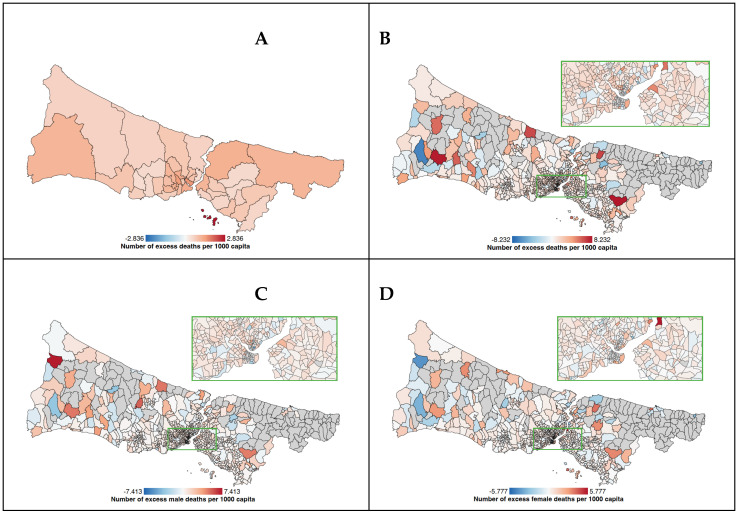
(**A**–**D**) Spatial distribution of EMR and selected variables at the district and neighborhood level in Istanbul. (**A**) District level EMR; (**B**) neighborhood-level EMR; (**C**) neighborhood-level male EMR; (**D**) neighborhood-level female EMR.

**Table 1 healthcare-12-01406-t001:** Characteristics of 808 neighborhoods with an over-1000 population in Istanbul based on selected variables (İstanbul, 2020).

Variable	Min.	1st Quartile	Median	Mean	3rd Quartile	Max.
Population density (per km^2^)	6.85	4689.59	15,838.07	21,240.07	31,947.52	95,270.13
Share of 50+ year old residents (%)	2.81	16.67	21.97	23.09	28.09	52.11
Share of residents with less than high school degree (%)	7.84	48.07	61.03	58.40	70.11	88.53
Share of residents with high school degree (%)	8.49	18.74	22.00	22.01	24.73	81.84
Share of residents with college or university degree (%)	2.69	10.46	16.08	19.59	25.79	60.51
SEVI score (0–100)	22.22	51.92	60.07	57.92	65.99	83.78
TVI score (0–100)	<0.001	17.17	19.71	20.49	22.78	79.15
Total EMR (per 1000)	−6.84	0.38	0.88	0.95	1.42	8.23
Female EMR (per 1000)	−4.34	0.02	0.33	0.37	0.64	5.78
Male EMR (per 1000)	−4.92	0.16	0.55	0.59	0.95	7.41
Total EMR (per 1000) in 10 neighborhoods with lowest SEVI scores	−0.79	-0.48	0.55	0.83	1.69	3.77
Total EMR (per 1000) in 10 neighborhoods with highest SEVI scores	−0.06	1.04	1.59	1.73	2.21	3.73

**Table 2 healthcare-12-01406-t002:** Simple linear regression results of the association between sociodemographic variables and EMRs according to gender in mortality data (Istanbul, 2020).

	Total		Female		Male	
Neighborhood Variables	Standardized Coefficient	*p*-Value	Standardized Coefficient	*p*-Value	Standardized Coefficient	*p*-Value
Population density	0.604	0.546	−0.383	0.701	1.169	0.243
Share of 50+ year old residents (%)	3.073	0.002	2.004	0.045	2.472	0.014
Share of residents with less than high school degree (%)	2.636	0.009	0.620	0.535	3.096	0.002
Share of residents with high school degree (%)	−1.654	0.099	−0.016	0.988	−2.272	0.023
Share of residents with college or university degree (%)	−2.589	0.010	−0.794	0.428	−2.877	0.004
SEVI Score	2.475	0.014	0.692	0.489	2.810	0.005
TVI Score	−2.725	0.007	−1.635	0.102	−2.321	0.021

**Table 3 healthcare-12-01406-t003:** Multivariable linear regression results of the association between selected sociodemographic variables and EMR according to gender in mortality data.

	Total		Female		Male	
Neighborhood Variables	Standardized Coefficient	*p*-Value	Standardized Coefficient	*p*-Value	Standardized Coefficient	*p*-Value
Population density	1.816	0.070	0.317	0.751	2.220	0.027
Share of 50+ year old residents (%)	4.865	<0.001	2.427	0.015	4.544	<0.001
SEVI Score	4.064	<0.001	1.527	0.127	4.241	<0.001
TVI Score	−1.329	0.184	0.944	0.345	−0.992	0.321

## Data Availability

The data that support the findings of this study are available from the Istanbul Metropolitan Municipality, but restrictions apply to the availability of these data, which were used under license for the current study and so are not publicly available. Data are, however, available from the authors upon reasonable request and with permission of the Istanbul Metropolitan Municipality.

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
