# Peer review of "Gender-Based Socio-Economic Inequalities in the Pre-Vaccination Era of the COVID-19 Pandemic in Istanbul: A Neighborhood-Level Analysis of Excess Mortality"

_healthcare, 2024, doi:10.3390/healthcare12141406_

Round 1
Reviewer 1 Report
Comments and Suggestions for Authors
Title and Abstract
1. The title includes "pre-vaccination," but it is absent from the abstract. Consider shortening the title for clarity.
2. It is advisable to incorporate the methods used into the abstract for completeness.
Introduction
3. Revise the introduction to align with the abstract and title, emphasizing the consideration of socioeconomic and sociodemographic factors, previous studies, and the methodology.
4. Specify the study's innovations.
5. Clarify how the study differs from previous research and contributes to international knowledge.
6. Ensure that the study's aims align with the title and abstract.
Materials and Methods
7. Describe the analytical methods employed, as the study lacks innovative or suitable spatio-temporal analysis techniques.
8. Establish a clearer link between the methods section, the title, and the abstract, particularly regarding the use of COVID-19 and pre-vaccination data, analysis methods, and the association between them.
Results, Discussion, and Conclusions
9. Provide data related to demographic and socioeconomic variables, including their basis and definition, as it is currently absent from the introduction and materials sections.
10. Explain the connection between the analysis and COVID-19, including the association with the pre-vaccination period mentioned in the title.
11. Ensure that the findings align with the study's title, specifically addressing gender disparities and socioeconomic inequalities.
12. Identify instances of socioeconomic inequality within the city.
13. Avoid exaggerated claims in the discussion section that exceed the scope of the analysis conducted.
14. Improve communication between sections and enhance the depth of analysis in the methods section to reflect innovation.
15. Ensure that the conclusions accurately reflect the study's findings.
Overall, address these points to enhance clarity, alignment, and depth in the manuscript. Thank you for your attention to these suggestions.
Author Response
Dear Reviewer 1
Please find our responses in the file attached.
Best regards

Reviewer 2 Report
Comments and Suggestions for Authors
I thank the authors for providing me the opportunity to review their interesting study on the direct and indirect effects of COVID-19 on mortality. The authors want to investigate the excess mortality rate (EMR) in Istanbul for 2020 and its association with selected demographic and socio-economic variables of development at the neighborhood level according to gender. Overall, the work is interesting, however, it presents some methodological weaknesses.
11) It is not very clear how the excess mortality from COVID-19 was assessed compared to the expected deaths that would have occurred if the pandemic had not occurred. Furthermore is not very clear how the authors calculated the excess mortality rate.
22) In the abstract it is reported that a logistic regression model was used (line 19), while in the methods section of the work the use of a linear regression model is declared (line 138; 168). Which of the two models was used? Reading the results section of the paper it seems that a linear regression model was used to evaluate EMR. Why the authors didn't use a Poisson regression?
33) Why did the authors decide to create three models to analyze the EMR and not a single model with sex as an adjustment covariate?
44) If the authors decide to accept the revision suggestions it would be appropriate to modify the results and possibly the discussions based on the new results.
Author Response
Dear Reviewer 2,
Please find our responses in the attached document.
Best regards

Round 2
Reviewer 1 Report
Comments and Suggestions for Authors
Thank you for revising the manuscript.
Author Response
We appreciate the comments provided by the reviewer that allowed us improve our manuscipt.
Reviewer 2 Report
Comments and Suggestions for Authors
Thank you for partially accepting my methodological suggestions, however the revised article now appears clearer.
Author Response
Thank your for your ccomments and suggestions. We appreciate your insight as it allowed us to clarify the cvertain issues in our manuscript.